# Hematological and Biochemical changes in *Schistosoma mansoni* infected patients at Haik Primary Hospital, North-East Ethiopia: A comparative cross-sectional study

**Habtye Bisetegn**[1]*, **Daniel Getacher Feleke**[2], **Habtu Debash**[1], **Yonas Erkihun**[1], **Hussen Ebrahim**[1]

1 Department of Medical Laboratory Sciences, College of Medicine and Health Sciences, Wollo University, Dessie, Ethiopia, 2 Department of Microbiology, Immunology and Parasitology, College of Health Sciences, Addis Ababa University, Addis Ababa, Ethiopia

* habtiye21@gmail.com

## Abstract

### Background

Schistosomes are blood dwelling parasites that affect more than 260 million people globally, and over 800 million people are at risk of infection in 74 countries. It causes acute and chronic debilitating diseases. The parasite is reported to alter the hematological and biochemical parameters in humans. Therefore, this study was aimed to evaluate the hematological and biochemical changes in *S. mansoni* infected adult patients compared to apparently healthy controls.

### Methods

A comparative cross-sectional study was conducted at Haik Primary Hospital from February to April 2021. One hundred and eighty study participants consisting of 90 *S. mansoni* infected patients and 90 apparently healthy controls were recruited using systematic random sampling method. Socio-demographic characteristics and other variables were collected using questionnaires. Stool sample was examined microscopically to detect *S. mansoni* infection using direct wet mount and Kato Katz technique. In apparently healthy controls, *S. mansoni* infection was rule out using direct wet mount and Kato Katz technique. Moreover, the intensity of *S. mansoni* infection was assessed using Kato Katz technique. Blood sample was collected from each study participant to determine the hematological and biochemical profiles. Data were entered in to Epi Data version 3.1 and analyzed using SPSS version 26.0 software. Kolmogorov-Smirnov and Shapiro Wilk normality tests were done to assess the distribution of continuous variables. The Mann-Whitney U test and Kruskal Wallis H test was done to compare the differences among nonnormally distributed variables between *S. mansoni* infected patients and healthy controls. P-values <0.05 at 95%CI were considered as statistically significant.

**Data Availability Statement:** All relevant data are within the manuscript and its Supporting Information files.

**Funding:** This research was funded by Wollo University, Northeast Ethiopia annual research grant. This grant was awarded to HB. The funders had no role in study design, data collection and analysis, decision to publish, or preparation of the manuscript. URL of funder:www.wu.edu.et.

**Competing interests:** The authors have declared that no competing interests exist.

## Result

The mean age (SD) of *S. mansoni* infected patients and apparently healthy controls was 30.33 (±12.26) and 31.2 (±12.85) years old, respectively. The prevalence of anemia, and thrombocytopenia among *S. mansoni* infected patients were 23.3% and 26.7%, respectively. Erythrocytic sedimentation rate (ESR) was significantly elevated among *S. mansoni* infected patients than apparently healthy controls. The median white blood cell count, red blood cell count, red blood cell indices, and platelet indices were significantly lower among *S. mansoni* infected patients compared to apparently healthy controls (P<0.05). On the other hand, the median eosinophil count was significantly elevated among *S. mansoni* infected patients compared to apparently healthy controls (P<0.05). This study also showed significantly elevated values of serum alanine aminotransferase, aspartate aminotransferase, alkaline phosphatase, and direct bilirubin and lower albumin, total cholesterol and triglycerides among *S. mansoni* infected patients compared to apparently healthy controls. Kruskal Wallis H test showed a significant difference in the median of most hematological and biochemical parameters between moderate and heavy intensity of infection with light intensity of infection and apparently healthy controls.

## Conclusion

The findings of this study showed significantly altered hematological values and liver function tests among *S. mansoni* infected patients compared to apparently healthy controls. Therefore, screening of *S. mansoni* infected patients for various hematological and biochemical parameters and providing treatment to the underlying abnormalities is very crucial to avoid schistosomiasis associated morbidity and mortality.

## Author summary

Schistosomiasis a significant parasitic disease that impose a major public health problem mainly in tropica and subtropical area of Africa. *Schistosoma mansoni* is endemic in different parts of Ethiopia including Haik town, and reported to be associated with significant morbidity and mortality in the community. The finding of the current study showed a significant change in hematological and biochemical parameters among *S. mansoni* infected adult patients. Anemia and thrombocytopenia are the most predominant hematological complications observed in patients with schistosomiasis. On the other hand, the intensity of infection is significantly corelated with the range of hematological abnormalities and alteration of liver function tests. Diagnosing *S. mansoni* infected patients for hematological and biochemical abnormalities is vital for controlling schistosomiasis associated morbidities and mortality. In addition, the finding of this study indicates the necessity of considering schistosomiasis as a cause of the abnormality or as a co-morbidity while interpreting hematological and biochemical profiles in schistosomiasis endemic areas.

## Introduction

Schistosomiasis is one of the most common neglected tropical disease caused by the blood dwelling trematodes of the genus *Schistosoma* [1]. Three Schistosoma species (*Schistosoma*

*mansoni*, *Schistosoma hematobium*, and *Schistosoma japonicum*) are responsible for the majority of Schistosomiasis cases in the World [2]. It is endemic in more than 74 countries with more than 260 million people infected and 800 million people being at risk of infection globally and of which, more than 90% of the cases occur in Africa [3,4]. Schistosomiasis is the second devastating parasitic disease following malaria [5]. Schistosomiasis is a major public health problem in sub-Saharan African countries with 120 million people infected and more than 150, 000 and 130, 000 died annually due to nonfunctioning kidney (from *S. haematobium*) and hematemesis (from *S. mansoni*), respectively [6,7].

*Schistosoma mansoni* infection is transmitted to humans through skin penetration by the freely swimming cercarial stage of the parasite during contact with contaminated water [8]. Several factors such as human population density, human contact with fresh water, presence or absence of piped water and sanitation facilities in the community, households, schools and health facilities, distance of snail habitat from human habitation, and snail control methods determine the persistent prevalence and intensity of schistosomiasis in a given ecological setting [2].

Clinically, Schistosomiasis results in acute, subacute and chronic clinical manifestations [9]. *Schistosoma mansoni* leads to gastrointestinal manifestations like nonspecific intermittent abdominal pain, diarrhea, and rectal bleeding, mucosal hyperplasia, pseudo-polyposis, and polyposis interspersed with normal bowel [10]. It also causes chronic hepatic schistosomiasis resulting from the host's granulomatous cell-mediated immune response to the soluble egg antigen of *S. mansoni*, which leads to irreversible fibrosis and severe portal hypertension [11]. Most of schistosomiasis morbidity and mortality are due to hepatic and intestinal granulomatous inflammation induced by the tissue trapped worm egg [12]. In patients with chronic and advanced schistosomiasis, elevated level of liver function tests such as aminotransferase (ALT), aspartate aminotransferase (AST), alkaline phosphatase (ALP), direct bilirubin (DBL), total bilirubin (TB) and declined level of white blood cell (WBC), red blood cells (RBC) and platelets were reported [13].

*Schistosoma mansoni* is an intravascular parasite that alters the hematological profiles and causes different hematological abnormalities such as anemia, thrombocytopenia, low RBC count, decreased level of RBC indices, and low WBC count [14–16]. A study conducted in Western Burkina Faso found significantly lower Hemoglobin (Hgb), hematocrit (HCT), mean cell volume (MCV), mean corpuscular hemoglobin concentration (MCHC) and mean platelet volume (MPV) among *S. mansoni* infected patients compared to apparently health controls [15].

Studies also reported the impact of *S. mansoni* infection on different liver function tests. A review on hepatosplenic Schistosomiasis reported that markers of liver injury such as ALT, AST, ALP, and bilirubin, were significantly higher among *S. mansoni* infected patients than healthy controls [17]. A study conducted in Bahia, Brazil reported that geometric means of liver injury biomarkers in *S. mansoni* infected women had significantly higher levels of ALT ($p < 0.01$), AST ($p < 0.01$), and γ-GT ($p < 0.01$) compared to those in uninfected controls [18]. A study conducted in Egypt reported a highly significant elevation of serum ALT, AST, and ALP in *S. mansoni* infected patients [19].

In Ethiopia, schistosomiasis is a major public health problem with more than 5 million people infected and more than 37 million people at risk of infection [20]. A systematic review and meta-analysis reported 18.7% (95%CI: 14.7–23.5) pooled prevalence of *S. mansoni* infection in Ethiopian general population and 37.13% (95%CI: 30.02–44.24) among children [21,22]. A study conducted in Northwest Ethiopia reported significantly lower mean values of Hgb, RBC, MCHC, total protein (TP), total cholesterol (TC), and total WBC count among *S. mansoni* infected patients compared to healthy controls. On the other hand, this study found an elevated values of AST and ALT among *S. mansoni* infected patients [23].

*Schistosoma mansoni* infection is reported to be endemic in Haik town [24]. However, there was no study conducted in the area to evaluate the impact of *S. mansoni* infection on hematological and biochemical profiles of patients. Therefore, this study was aimed to assess the impact *of S. mansoni* infection on hematological and biochemical changes among adult *S. mansoni* patients attending Haik Primary Hospital, Haik town, North-East Ethiopia.

## Methods and materials

### Ethical approval and consent to participate

Ethical approval was obtained from the Research and Ethical Review Committee of the Wollo University, College of Medicine and Health Sciences with ethical review number "CMHS190/02/2021". Written informed consent was obtained from each study participant. All information pertaining to patient biodata that could be potentially used to identify the patient such as the patient names, were converted in to codes. Access to the original patient data was allowed only to the principal investigator and to the hospitals' medical laboratory technologists. Study participants with confirmed *S. mansoni* infection, underlying hematological, biochemical and platelet abnormalities were treated according to the hospital treatment guideline.

### Study area

This study was conducted at Haik Primary Hospital. Haik Primary Hospital is found in Haik town, South Wollo Zone, Amhara Regional State, North-East Ethiopia, about 430 Km far from Addis Ababa, the capital city of Ethiopia (Fig 1). The town has a geographic coordinate of 13030.59"N latitude and 039028.849"E longitude. Its altitude is about 2200m above sea level

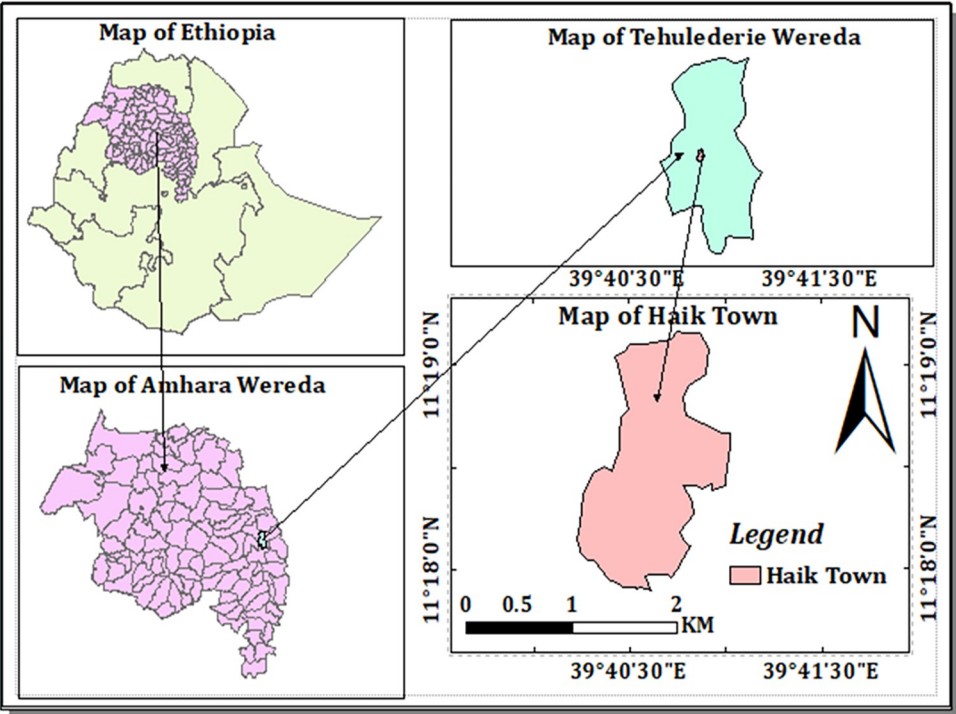

**Fig 1. Map of the study area.** The above maps were generated by ourselves using the ArcGIS version 10.8 software. The shape files of Ethiopia administrative regions, zones, woredas and towns were freely downloaded from a link https://africaopendata.org/dataset/ethiopia-shapefiles.

and the area covers 447.8 $km^2$ with a total population of 108,993. The major occupations of the inhabitants include trades, civil service, daily labor, and subsistence agriculture in the suburban villages. There are different water bodies in the district like Ankerca River, Logo Lake, Ardibo Lake, and Kette River which are used as major sources of irrigation activities and sanitation in the district. These rivers might be served as a habitat for the snail intermediate host of *S. mansoni*. Previously conducted studies indicated that *S. mansoni* is a public health problem and endemic in Haik town. A study done among school children reported the prevalence of *S. mansoni* infection to be 45%. [25].

## Study design and period

Health facility based comparative cross-sectional study was conducted at Haik Primary Hospital North-East Ethiopia, from February to April 2021.

## Eligibility criteria

In this study, all *S. mansoni* infected adults (≥18 years old) visiting Haik Primary Hospital during the study period and who are willing to participate in the study were included. Apparently healthy controls (participants, who are negative for *S. mansoni* and other parasitic infections that are reported to alter the hematological and biochemical parameters of infected individuals) were recruited from healthy blood donors while donating blood at Dessie blood bank. However, Chronic alcohol drinkers, regular khat chewer, current cigarette smokers, viral hepatitis B and C patients, HIV/ADIS patients, pregnant women, anticoagulant or anti-aggregant drug users and chronic diseases apart from Schistosomiasis were carefully excluded. Patients were also excluded if they are presented with splenectomy, diabetes mellitus, hypertension, use hepatotoxic drugs, thrombocytopenic drugs, or drugs that change platelet function (such as acetylsalicylic acid), and chronic renal disease. Lactating mothers, and Patients with a history of malignancy were also excluded. Patients infected with other parasitic diseases which cause alteration of hematological and biochemical profiles were excluded. Both *S. mansoni* infected patients and healthy controls were screened for malaria, Hookworm infection, and tirichuriasis and those infected with these parasites were excluded.

## Sample size and sampling technique

According to the rule of thumb recommended by van Voorhis and Morgan, 30 participants per group are required to detect real differences which could lead to 80% power [26]). In this study, more study participants have been recruited to increase the reliability of the study. Thus, a total of 180 study participants (90 *S. mansoni* infected adults and 90 apparently healthy controls) were enrolled in this study. A systematic random sampling technique was employed to recruit the study participants at Haik Primary Hospital based on the inclusion and exclusion criteria. While reviewing the hospital registration book, on average two *S. mansoni* infected adult are expected to be reported in the Hospital. Within 90 days of data collection period a total of 180 cases are expected to be reported. Then K value was calculated as 180/90 = 2. Finally, with lottery method sampling was begin in the second S. mansoni infected patient and the sampling process continued by systematically recruiting every other patient.

## Data collection and Laboratory investigation

**Socio-demographic characteristics and clinical data.** Socio-demographic characteristics and other variables about the study participants were collected using pre-tested questionnaire by the investigators and trained data collectors. Clinical information and physical diagnosis of

*S. mansoni* infected adults and apparently healthy controls were assessed by physicians and trained nurses at Haik Primary Hospital and Dessie blood bank, respectively.

**Stool sample collection and processing.** Stool sample was collected from each *S. mansoni* suspected patient visiting Haik Primary Hospital and healthy control at Dessie blood bank by a leak-proof container. *Schistosoma mansoni* was diagnosed by direct wet mount using normal saline (0.85% sodium chloride solution) and Kato–Katz technique at Haik Primary Hospital laboratory. Proportion of the stool sample of patients with confirmed *S. mansoni* infection was preserved with formalin and transported to Wollo university laboratory to determine the intensity of infection using the Kato–Katz technique.

**Blood sample collection and processing.** Following an aseptic technique, about 6 ml of venous blood was collected using a sterile disposable syringe from each study participant by experienced blood sample collectors at Haik primary hospital and Dessie blood bank. Then, about 3 ml of the collected blood was added into a EDTA anticoagulated test tube and used for complete blood count (CBC), while the remaining 3 ml of blood was dispensed into a serum separator tube for the biochemical test as described elsewhere [27,28].

**Determination of hematological and biochemical profiles.** Complete blood count was done using DIRUI BF 6500 automated hematology analyzer (Dirui Industrial CO. Ltd., P.R., China) within 2 hours of blood collection. The instrument works based on the principle of semiconductor laser flow cytometry combined with cytochemical staining, electrical impedance, and cyanide free colorimetry to characterize blood cells and hemoglobin (Hgb) according to Hgb cyanide free Hgb measurement method as described elsewhere [27]. The hematological parameters were classified as low, normal and high based on the established hematological reference range [29].

The liver function tests such as aspartate and alanine aminotransferases (AST and ALT), total bilirubin (TB), direct bilirubin (DB), total protein (TP), albumin protein (ALB), and alkaline phosphatase (ALP) were measured by DIRUI CS-T240 fully automated biochemistry analyzer (Dirui Industrial Co., Ltd. India). The results of the biochemical profiles were classified as low, normal, and high based on the locally established reference range for selected clinical chemistry parameters in Northern Ethiopia [28].

**Statistical analysis.** Data were entered to Epi Data version 3.1 and exported to statistical package for social science (SPSS) version 26.0 software for statistical analysis. Socio-demographic and other variables were presented in the form of frequency and percentage by tables. Kolmogorov-Smirnov and Shapiro Wilk normality tests were conducted to check the distribution of continuous variables. The continuous variables were s not normally distributed and expressed in the form of the median (interquartile range, IQR). Since the continuous variables were not normally distributed and the groups to be compared were four, the non-parametric Kruskal Wallis H test was used to compare the hematological and biochemical parameters between patients with light, moderate and heavy intensity of infection and health controls. On the other hand, the Mann-Whitney U test was used to compare differences among non-normally distributed variables between *S. mansoni* infected and non-infected study participants. Variables with a P-value of <0.05 at 95%CI were considered as statistically significant.

**Quality assurance.** The quality of the blood sample was maintained by collecting and processing it according to the standard operating procedures (SOPs) [27,30]. Samples were checked to the acceptable criteria like the absence of hemolysis and clotting, sample volume, collection time, and correct labeling. Safety and specimen handling procedures were strictly followed. The performance of the coagulometer was checked by the daily running of two-level controls (Normal and High). The performance of the hematology analyzer was checked by daily background checking. The quality of the Kato Katz was checked daily and 10% of the

slides were randomly selected and reexamined by an experienced laboratory technologist who was blind to the first examination result.

## Result

### Socio-demographic characteristics of the study participants

In this study, a total of 180 study participants (90 *S. mansoni* infected adults consisting of 48 males and 42 females with age between 18 and 62 years old and 90 healthy controls consisting 49 males and 41 females with age between 19 and 54 years old) were involved. The mean age (SD) of *S. mansoni* infected participants and healthy controls was 30.33±12.26 and 31.2 ±12.85 years old, respectively. About 60% of *S. mansoni* infected study participants are urban residents and 48.9% of them were married. About 30% of the study participants were illiterate (Table 1).

### Hematological profiles of *S. mansoni* infected patients

Out of 90 *S. mansoni* infected study participants, 33.3% of them had high WBC count and 26.7% of them had eosinophilia. On the other hand, the majority of the *S. mansoni* infected patients had normal basophil count (93.3%), monocyte count (83.3%), and lymphocyte counts (76.7%). Almost all of the *S. mansoni* infected study participants had low MCH (96.1%) and RDW (90%). About 26.7% of the study participants had low MCV. Thrombocytopenia was found in 26.7% of *S. mansoni* infected study participants (Table 2).

### Prevalence and severity of anemia among *S. mansoni* infected patients

The overall prevalence of anemia among *S. mansoni* infected patients was 23.3%. The prevalence was higher among females (28.6%) than among males (18.8%). Regarding severity, about 12.5% and 6.25% of males had mild and moderate anemia, respectively. Moreover, 21.4% and

**Table 1. Socio-demographic characteristics, of *S. mansoni* infected study participants and healthy controls at Haik Primary Hospital, North-East Ethiopia, from February to April 2021.**

| Variable | | *S. mansoni* positive (n = 90) N (%) | Control (n = 90) N (%) |
|---|---|---|---|
| Mean age and standard division | | 30.33±12.26 | 31.2 ±12.85 |
| Sex | Male | 48 (53.3) | 49 (54.4) |
| | Female | 42 (46.7) | 41(45.6) |
| Residence | Urban | 54 (60) | 50 (55.6) |
| | Rural | 36 (40) | 40 (44.4) |
| Marital status | Single | 43 (47.8) | 40(44.4) |
| | Married | 44 (48.9) | 46(51.1) |
| | Divorce | 0 | 4(4.4) |
| | Widowed | 3 (3.3) | 0 |
| Educational status | Illiterate | 27 (30) | 28(31.1) |
| | Primary | 24 (26.7) | 22(24.4) |
| | Secondary | 24 (26.7) | 23(25.6) |
| | College and above | 15 (19.7) | 17(16.7) |
| Occupation | Student | 27 (30) | 27(30) |
| | Gov.t employee | 9(10) | 11(12.2) |
| | Farmer | 12 (13.3) | 13(14.4) |
| | House wife | 27(30) | 25(27.8) |
| | Private | 15(19.7) | 16(17.8) |

**Table 2. Hematological profiles of *S. mansoni* infected study participants visiting Haik Primary Hospital, North-East Ethiopia, from February to April 2021.**

| Variable | S. mansoni infected adult (n = 90) | | | | |
|---|---|---|---|---|---|
| | Normal n (%) | Low n (%) | High n (%) | Reference range | |
| | | | | Male | Female |
| WBC ($10^3$µL) | 15(50) | 45(16.3) | 30(33.3) | 3.6–10.6 | 3.6–10.6 |
| Neutrophile ($10^3$µL) | 51(56.7) | 21(23.3) | 18(20) | 1.7–7.5 | 1.7–7.5 |
| Lymphocyte ($10^3$µL) | 69(76.7) | 18(20) | 3(3.3) | 1.0–3.2 | 1.0–3.2 |
| Monocyte ($10^3$µL) | 75(83.3) | 12(13.3) | 3(3.3) | 0.1–1.3 | 0.1–1.3 |
| Eosinophile ($10^3$µL) | 66(73.3) | - | 24(26.7) | 0–0.3 | 0–0.3 |
| Basophil($10^3$µL) | 84(93.3) | - | 6(6.7) | 0–0.2 | 0–0.2 |
| RBC ($10^6$µL) | 19(21.1) | 71(78.9) | - | 4.2–6.0 | 3.8–5.2 |
| HCT (%) | 60(66.7) | 12(13.3) | 18(20) | 40–54 | 35–49 |
| MCV (Fl) | 66(73.3) | 24(26.7) | - | 80–100 | 80–100 |
| MCH (pg) | 3(3.3) | 87(96.7) | - | 26–34 | 26–34 |
| MCHC (g/Dl) | 39(43.3) | 51(56.7) | - | 32–36 | 32–36 |
| RDW (%) | 9(10) | 81(90) | - | 11.5–14.5 | 11.5–14.5 |
| PT ($10^3$µL) | 60(66.7) | 24(26.7) | 6(6.7) | 150–450 | 150–450 |
| MPV (Fl) | 75(83.3) | 15(16.7) | - | 7.0–12.0 | 7.0–12.0 |
| PDW (%) | 42(46.7) | 9(10) | 39(43.3) | 9.6–16 | 9.6–16 |
| ESR (mm/h) | 72(80%) | 18(20%) | - | 0–15 | 0–20 |

WBC: White blood cell; RBC: Red blood cell; PT: Platelet; Hgb: Hemoglobin; HCT: Hematocrit; MCV: Mean cell volume; RDW: Red cell distribution width; MCHC: Mean corpuscular hemoglobin concentration; MPV: Mean platelet volume; PDW: Platelet distribution width; ESR: Erythrocytic sedimentation rate.

7.14% of females had mild and moderate anemia (Fig 2). In majority of *S. mansoni* infected patients, the type of anemia observed was microcytic hypochromic that corresponds to low level of MCV and MCHC.

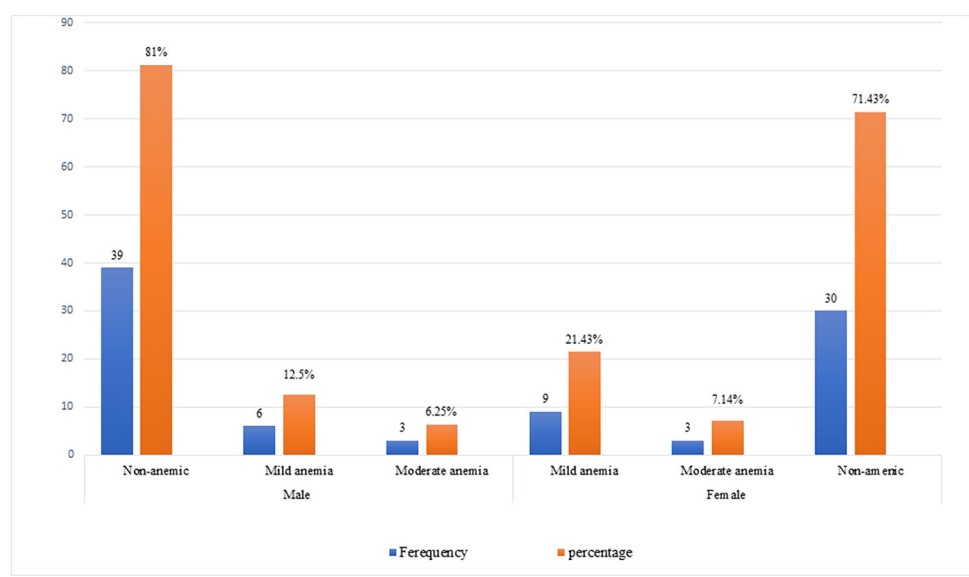

**Fig 2. Severity of anemia among *S. mansoni* infected patients: The first chart indicate the severity amnemia among males while the last three chart indicate anemia severity in females.** The blue chart indicate the frequency and the red indicate the percent of anemia.

**Table 3. Biochemical profiles of *S. mansoni* infected adults and healthy controls at Haik Primary Hospital, North-East Ethiopia, from February to April 2021.**

| variables | | *S. mansoni* infected adults (N = 90) | Healthy controls (N = 90) | Reference range |
|---|---|---|---|---|
| | | Frequency (%) | Frequency (%) | |
| ALT (IU/L) | Normal | 66 (73.3) | 85 (94.4) | 5–23 |
| | High | 24(26.7) | 5(5.6) | |
| AST (IU/L) | Normal | 8(8.9) | 87(96.7) | 14.2–34.9 |
| | High | 82(91.1) | 3(3.3) | |
| TP (g/dL) | Low | 6(6.6) | 0 | 6.09–7.85 |
| | Normal | 84(93.3) | 90(100) | |
| | High | 0 | 0 | |
| ALB (g/dL) | Low | 14(15.6) | 1(1.1) | 4.42–5.46 |
| | Normal | 76(84.4) | 89(98.9) | |
| | High | 0 | 0 | |
| TBIL (mg/dL) | Normal | 87(96) | 90(100) | 0.1–0.81 |
| | High | 3(3.3) | 0 | |
| DBIL (mg/dL) | Normal | 39(43.3) | 88(97.8) | 0.03–0.53 |
| | High | 51(56.7) | 2(2.2) | |
| ALP (IU/L) | Low | 0 | 0 | 66–456 |
| | Normal | 9(10) | 88(97.8) | |
| | High | 81(90) | 2(2.2) | |
| TG (mg/dL) | Normal | 90 | 90(100) | <150 (Fasting) |
| | High | 0 | 0 | ≥150 (Fasting) |
| TC (mg/dL) | Normal | 90 | 90(100) | <150 |

ALT: Alanine aminotransferase; ALP: Alkaline phosphatase; AST: Aspartate aminotransferase; L: liter; mg: Milligram; dL: deciliter; IU: International Unit; TP: Total protein; ALB: Albumin; DBIL: Direct Bilirubin; TBIL: Total Bilirubin; TC: Total cholesterol; TG: Triglyceride.

## Biochemical profiles of *S. mansoni* infected adults and healthy controls

The levels of ALT (26.7%), AST (91.1%), DBIL (56.7%), and ALP (90%) were increased in *S. mansoni* infected patients. However, 15.6% and 6.6% of *S. mansoni* infected patients had low level of albumin and total protein, respectively. The level of TC and TG in *S. mansoni* infected patients and healthy controls were within the normal reference intervals. Almost all biochemical profiles in apparently healthy controls were within the normal reference range (Table 3).

## Comparison of hematological and biochemical profiles between *S. mansoni* infected patients and apparently healthy controls

*Schistosoma mansoni* infected patients had significantly lower hematological values compared to the healthy controls (P <0.05). However, Eosinophile count was significantly higher among *S. mansoni* infected patients compared to healthy controls with a median [IQR] of 0.37 (0.29) $\times 10^3 \mu L$ and 0.15(0.3) $\times 10^3 \mu L$, respectively (P<0.001). The Mann-Whitney U test showed significantly lower WBC count, RBC count, Hgb value, MCV, MCH, MCHC, RDW, MPV, PDW and ESR in *S. mansoni* infected adults compared to healthy controls (p<0.05) (Table 4).

The Mann-Whitney U test also showed significantly elevated values for AST, ALT, DBIL and ALP in *S. mansoni* infected patients compared to the control group (P-value 0.05). On the other hand, the TC, ALB and TG level were significantly decreased among *S. mansoni* infected patients compared to apparently healthy controls (P-value <0.05) (Table 4).

**Table 4. Comparison of hematological and biochemical profiles among *S. mansoni* infected adults and apparently healthy controls at Haik Primary Hospital, North-East Ethiopia from February to April 2021.**

| Variable | *S. mansoni* infected adult (n = 90) | | Healthy controls (n = 90) | | P-value |
|---|---|---|---|---|---|
| | Median (IQR) | 95%CI | Median (IQR) | 95%CI | |
| WBC ($10^3$µL) | 5.8 (3.9) | 3.01–8.24 | 7.8 (4.8) | 7.00–9.38 | 0.032 |
| Neutrophile ($10^3$µL) | 3.67 (5.2) | 2.98–3.45 | 3.94(3.2) | 3.89–4.02 | 0.052 |
| Lymphocyte ($10^3$µL) | 2.1 (1) | 1.97–2.3 | 2.24(1.1) | 2.05–2.35 | 0.046 |
| Monocyte ($10^3$µL) | 0.44 (0.26) | 0.36–0.46 | 0.51(0.3) | 0.48–0.61 | 0.061 |
| Eosinophile ($10^3$µL) | 0.37 (0.29) | 0.29–0.42 | 0.15(0.3) | 0.098–0.174 | 0.001 |
| Basophil($10^3$µL) | 0.065 (.04) | 0.055–0.07 | 0.064(0.03) | 0.061–0.07 | 0.91 |
| RBC ($10^6$µL) | 3.8 (1.13) | 3.12–4.02 | 5.8(1.13) | 5.31–5.99 | 0.012 |
| Hgb (g/dL) | 14.2 (3.1) | 10.64–18.24 | 15.42(6.4) | 12.84–19.21 | 0.024 |
| HCT (%) | 42.6 (6.42) | 33.51–45.98 | 46.31(9.4) | 39.19–52.01 | 0.061 |
| MCV (fL) | 81.7 (7.9) | 77.95–82.15 | 86.57(8.4) | 80.97–90.00 | < 0.001 |
| MCH (pg) | 24.6 (2.6) | 24.2–25 | 28.82(3.42) | 26.49–29.21 | < 0.001 |
| MCHC (g/dL) | 30 (0.9) | 26.85–30.24 | 33.4(2.6) | 31.74–33.85 | < 0.001 |
| RDW (fL) | 37.7 (6.4) | 36.35–38.7 | 62.88(12.3) | 59.41–64.20 | < 0.001 |
| PT ($10^3$µL) | 244 (152) | 230.5–258.5 | 277.0 (75) | 262.0–279.32 | 0.215 |
| MPV (fL) | 6.05 (1.6) | 5.55–6.4 | 15.6 (1.0) | 14.63–16.01 | < 0.001 |
| PDW (fL) | 17.6 (3.5) | 17.1–18.4 | 9.7 (1.2) | 9.41–9.89 | < 0.001 |
| ESR (mm) | 15.3(5.8) | 12.86–15.96 | 5.4(1.1) | 4.87–5.76 | <0.001 |
| ALT (IU/L) | 28(19) | 26.00–32.95 | 20.23(16.21) | 19.04–21.00 | <0.04 |
| AST (IU/L) | 55.5(23) | 52.5–60.00 | 24.41(17.64) | 21.51–26.01 | ≤0.001 |
| TP (g/dL) | 7.35(1.1) | 7.15–7.55 | 7.83(1.6) | 7.57–9.12 | 0.215 |
| ALB (g/dL) | 3.25(0.7) | 3.01–3.42 | 4.47(0.84) | 3.85–4.73 | 0.036 |
| TBIL (mg/dL) | 0.44(0.3) | 0.32–0.56 | 0.43(0.2) | 0.31–0.48 | 0.45 |
| DBIL (mg/dL) | 0.26(0.18) | 0.195–0.28 | 0.12(0.1) | 0.098–0.25 | 0.02 |
| ALP(IU/L) | 425.3(355) | 382.95–459.80 | 128(55.3) | 118.07–156.91 | ≤0.001 |
| TG (mg/dL) | 31.6(18.6) | 29.45–34.90 | 86.7(21.8) | 52.71–94.56 | 0.026 |
| TC (mg/dL) | 82.75(21.8) | 76.35–88.85 | 121(32.1) | 115.98–142.58 | 0.004 |

## Intensity of infection, biochemical and hematological values

The overall median EPG of *S. mansoni* infected patients was 215 EPG. Of the 90 *S. mansoni* infected patients, 55.6% (95%CI: 44.4–65.6), 28.9% (95%CI: 20–38.9), and 15.6% (95%CI, 8.9–23.3) had light, moderate, and heavy intensity of infection respectively. The WBC and RBC count was significantly lower in patients with moderate and heavy intensity of infection compared to patients with light intensity of infection and apparently healthy controls (P<0.05). The value of Hgb decreases as the intensity of infection increase. However, this difference was not statistically significant (P>0.05). According to the Kruskal Wallis H test, the median WBC count, RBC count, MCV, RDW, MCHC, MCH, MPV, PDW, Hgb and ALB in patients with moderate and heavy intensity of infection were significantly lower than in patients with light intensity of infection, and in healthy controls (P<0.05). On the other hand, the values of AST, ALT, DBIL and ALP were significantly elevated among patients with moderate, and heavy intensity of infection compared to those with light intensity of infection, and healthy controls (P<0.05) (Table 5).

## Discussion

*Schistosoma mansoni* is a blood-dwelling parasitic worm that alters the hematological and biochemical profiles in patients with *S. mansoni* infection [31]. The finding of this study showed

**Table 5. Association of S. mansoni intensity of infection with hematological and biochemical profiles of patients and healthy controls.**

| Variable | Light (1–99 EPG) | Moderate (100–399 EPG) | Heavy (>400 EPG) | Healthy control (0 EPG) | P-value |
|---|---|---|---|---|---|
| | Median (IQR) | Median (IQR) | Median (IQR) | Median (IQR) | |
| WBC ($10^3$μL) | 7.69 (4.21) | 3.42 (5.2) | 2.19 (3.14) | 7.8 (4.8) | 0.001 |
| Neutrophile ($10^3$μL) | 3.74 (5.3) | 3.21(2.1) | 2.84 (1.8) | 3.94(3.2) | 0.21 |
| Lymphocyte ($10^3$μL) | 2.4 (0.98) | 1.80 (0.55) | 1.01 (0.42) | 2.24(1.1) | 0.08 |
| Monocyte ($10^3$μL) | 0.46 (0.18) | 0.42 (0.1) | 0.38 (0.09) | 0.51(0.3) | 0.16 |
| Eosinophile ($10^3$μL) | 0.32 (0.098) | 0.38 (0.12) | 0.43 (0.23) | 0.15(0.3) | 0.001 |
| Basophil($10^3$μL) | 0.067 (0.04) | 0.064 (0.04) | 0.0635 (0.04) | 0.064(0.03) | 0.7 |
| RBC ($10^6$μL) | 4.02 (3.2) | 3.61(2.41) | 2.05 (1.02) | 5.8(1.13) | <0.001 |
| Hgb (g/dL) | 16.58 (8.7) | 13.21 (6.32) | 11.63 (5.84) | 15.42(6.4) | <0.001 |
| HCT (%) | 44.3 (22.3) | 38.27 (22) | 35.64 (21.4) | 46.31(9.4) | <0.001 |
| MCV (fL) | 82.01 (18.6) | 76.42 (18.3) | 74.51 (18.01) | 86.57(8.4) | < 0.001 |
| MCH (pg) | 26.25 (5.4) | 24.03 (5.4) | 22.38 (5.34) | 28.82(3.42) | < 0.001 |
| MCHC (g/dL) | 31.71 (1.01) | 29.31(1.2) | 27.98 (0.98) | 33.4(2.6) | < 0.001 |
| RDW (fL) | 38.81 (7.1) | 37.02 (6.83) | 36.31(6.84) | 62.88(12.3) | < 0.01 |
| PT ($10^3$μL) | 252 (144) | 238 (123) | 236 (122.6) | 277.0 (75) | 0.051 |
| MPV (fL) | 6.39 (1.65) | 5.81 (1.61) | 5.49 (1.61) | 15.6 (1.0) | < 0.001 |
| PDW (fL) | 18.39 (4.14) | 17.4 (3.4) | 17.01 (3.35) | 9.7 (1.2) | < 0.001 |
| ESR (mm) | 16.01(5.98) | 14.21 (5.01) | 12.93 (3.17) | 5.4(1.1) | <0.001 |
| ALT (IU/L) | 24.9 (18.02) | 28.1 (19.01) | 32.79 (20.3) | 20.23(16.21) | <0.04 |
| AST (IU/L) | 46.44 (21.53) | 49.23 (22.01) | 59.37 (24.2) | 24.41(17.64) | <0.001 |
| TP (g/dL) | 7.49 (1.03) | 7.25 (1.01) | 7.03 (0.98) | 7.83(1.6) | 0.06 |
| ALB (g/dL) | 3.45 (0.91) | 3.14 (0.71) | 3.00 (0.65) | 4.47(0.84) | 0.04 |
| TBIL (mg/dL) | 0.33 (0.023) | 0.64 (0.1) | 0.51 (0.14) | 0.43(0.2) | 0.21 |
| DBIL (mg/dL) | 0.15 (0.03) | 0.24 (0.12) | 0.28 (016) | 0.12(0.1) | 0.03 |
| ALP(IU/L) | 381.9 (265) | 431.3 (344) | 438.9 (356) | 128(55.3) | <0.001 |
| TG (mg/dL) | 29.84 (16.2) | 31.23(18.3) | 33.21(18.6) | 86.7(21.8) | 0.045 |
| TC (mg/dL) | 80.49 (18.1) | 82.35 (21.75) | 84.19 (22.01) | 121(32.1) | 0.04 |

EPG: egg per-gram of stool

that the median of WBC count was significantly lower in *S. mansoni* infected adult patients compared to apparently healthy controls ($5.8 \times 10^3$μL vs $7.8 \times 10^3$μL) (P<0.032). This finding was supported by report from Sanja Town Northwest Ethiopia [23]. In addition, this finding was also supported by a study that reported decreased WBC count in 57% of *S. mansoni* infected patients [32]. The Eosinophil count was significantly higher among *S. mansoni* infected adults than in healthy controls. This was in agreement with a report from Brazil and Western Uganda [33,34]. Eosinophilia in *S. mansoni* infected patients is due to powerful defense reactions against the tissue migrating larvae during acute phase of infection, release of antigen from spontaneously dying parasites or after chemotherapy, activation of proteolytic enzymes produced by the egg, and allergic manifestation [34,35]. However, the value of neutrophiles, lymphocytes, basophils, and monocytes were not significantly different between *S. mansoni* infected patients and apparently healthy controls (P>0.05). This finding agreed with a report of experimental studies conducted in Alexandria Egypt, and Gabon [36,37].

The median RBC count, Hgb level, and the value of RBC indices such as RDW, MCV, MCHC, and MCH were significantly lower among *S. mansoni* infected patients compared to healthy controls (P<0.001). This finding was in line with reports from Northwest Ethiopia [23] and Western Burkina Faso [15]. In this study, there was no significant difference in HCT values between *S. mansoni* infected adults and healthy controls (P<0.062).

The median platelet count was lower in *S. mansoni* infected patients ($244 \times 10^3$) compared to the healthy controls ($277 \times 10^3$). However, this difference was not statistically significant ($P > 0.05$). This finding disagrees with a study done in Brazil, where the difference was statistically significant [16]. In this study, 26.7% of *S. mansoni* infected patients were thrombocytopenic. The mechanism of thrombocytopenia in *S. mansoni* infected patients is reported to be multifactorial. According to the finding of different studies, the most common cause of thrombocytopenia among *S. mansoni* infected patients are gastrointestinal bleeding, massive adhesion of platelets to the *S. mansoni* egg shell in the vascular endothelium, and *S. mansoni* infected patients producing anti-schistosome antibodies that cross-react with platelet and this leads to clearance of the platelet and cause thrombocytopenia [38,39]. In addition, splenomegaly caused by congestion due to obstruction, hyperplasia of reticulo-endothelial cells, fibrosis due to immunological stimulation, and hypersplenism were reported to cause thrombocytopenia [12,39,40]. Patients with chronic hepatosplenic schistosomiasis are more risky to develop complex hemostatic abnormality, that leads to risk of life threatening bleeding from ruptured esophageal varices [41].

Erythrocytic sedimentation rate (ESR) was significantly higher in *S. mansoni* infected patients than healthy controls. This finding was agreed with reports from Nigeria [32]. The elevation of ESR in *S. mansoni* patients might be due to blood loss as a result of bleeding during migration of worms in the intestine and consumption of blood by the worm.

In the present study, the overall prevalence of anemia among *S. mansoni* infected patients was 23.3%. This finding was in line with a study done in Kenya that reported 25.5% (13.1–38.0) prevalence of anemia among patients with heavy intensity of *S. mansoni* infection [42]. However, the finding was lower than a study conducted in Brazil [40]. The difference might be due to variation in immune status of the study participants, difference in strains of the parasite, and the intensity of infection. The prevalence of anemia among males and females was 18.8% and 28.6%, respectively. Regarding the severity, about 12.5% and 6.25% of males had mild and moderate anemia, respectively. Of the anemic female participants, 21.4% of them were mild anemic and 7.14% were moderately anemic. Anemia is one of the hematological abnormalities in *S. mansoni* infected patients that contribute to schistosomiasis associated morbidity and mortality [43]. The possible mechanisms of anemia in *S. mansoni* infected patients are iron deficiency due to extra-corporeal blood loss from egg movement through the intestinal wall, red blood cells, sequestration due to splenomegaly, anemia of inflammation, autoimmune hemolysis, gastrointestinal bleeding due to movement of laterally spined S. mansoni egg and anemia of inflammation induced by proinflamatory cytokines [44,45]. Schistosomiasis associated anemia is also contribute to fatigue, weakness, reduced cognitive function and impose socioeconomic burden [45].

*Schistosoma mansoni* infected patients were found to have elevated levels of ALT, AST, DBIL, and ALP. These parameters are important marker of liver injury in patients with schistosomiasis. This finding was in line with a study conducted in Northwest Ethiopia that reported a significantly elevated amounts of AST, and ALT among *S. mansoni* infected patients compared to apparently healthy controls [23] and other studies reported significantly higher AST, ALT, ALP and bilirubin among patients with hepatosplenic schistosomiasis than healthy controls [17–19,46]. The elevated value of liver function tests in *S. mansoni* infected patients might be due to the impairment of liver function due to *S. mansoni* egg deposition in the liver that later induces early granuloma formation and later on portal fibrosis and enlarged fibrotic portal tract [11]. The total protein and albumin levels were lower in 3.3% and 10% of the *S. mansoni* infected patients, respectively. This finding was supported by a study in Brazil that reported significantly lower levels of albumin in *S. mansoni* infected patients compared to the healthy controls [17]. In addition, the level of TC and TG was significantly lower in *S. mansoni*

infected adults than the noninfected one. This finding was supported by the finding of a study conducted in Brazil [47]. The low level of cholesterol in schistosomiasis patients is reported to be associated with the parasite need but cannot synthesize cholesterol. So, the explanation is the adult worm internalize the host cholesterol through their tegument, shedding of Glycosyl-phosphatidylinositol sequestered with the host lipoprotein, and lipoprotein removal by neutrophil endocytosis [48,49].

The non-parametric Mann-Whitney U tests showed a significant difference in ALT, AST, DBIL, and ALP between *S. mansoni* infected patients and healthy controls (P-value <0.04). This was supported by studies conducted in Brazil and Northwest Ethiopia [23,50].

The findings of this study showed 55.6%, 28.9%, and 15.6% light, moderate, and heavy intensity of *S. mansoni* infection among the study participants, 'respectively. This finding was in line with a report from Southwest Ethiopia that reported 57% light, 26.7% moderate, and 16.3% heavy intensity of infection [51]. However, the finding was quite different from reports in Sanja town, Northwest Ethiopia, that reported 69.1% light, 28.2% moderate, and 2.7% heavy intensity of infection [23] and Sanja Primary hospital Northwest Ethiopia that reported 40.0% light, 24.0% moderate and 36.0% heavy intensity of infection respectively [52]. According to the Kruskal Wallis H test, the median WBC count, RBC count, MCV, RDW, MCHC, MCH, MPV, PDW, and ALB in patients with moderate and heavy intensity of infection was significantly lower than in patients with light intensity of infection and healthy controls (P<0.05). on the other hand, the values of AST, ALT, DBIL, and ALP were significantly elevated among patients with Moderate and heavy intensity of infection compared to those with light intensity of infection and healthy controls (P<0.05).

This study was conducted with some limitations. Imaging techniques such as magnetic resonance angiography (MRA) and magnetic resonance imaging (MRI) was not done to assess to the morphological alteration of the liver and spleen due to schistosomiasis. The disease clinical severity parameters like size of the spleen, size of liver sign of fibrosis, and thickness of portal vein at entrance and secondary branch and its association with the various hematological and biochemical parameters were not evaluated. In addition, chronicity of the disease, and the markers of fibrosis and granulomatous inflammatory cytokine were not determined. Further study addressing these limitations are recommended to be conducted. As anemia is the leading hematological abnormalities, further studies that can investigate the mechanism and type of anemia among patients with schistosomiasis is also recommended.

## Conclusion

The findings of this study showed significantly altered hematological and biochemical parameters among *S. mansoni* infected patients compared to apparently healthy controls. Considering hematological and biochemical abnormalities in schistosomiasis patients, and screening of patients with hematological and/or biochemical abnormalities for S. mansoni infection in endemic areas is an important measure to reduce schistosomiasis associated morbidity and mortality as well as to improve the health of the society. In addition, anemia and thrombocytopenia are found to be a major health problem among *S. mansoni* infected patients. As most of the hematological parameters declined with increased intensity of *S. mansoni* infection, treatment of the infection and the underlying hematological and biochemical abnormalities is recommended. Considering the range of infection intensity and associated clinical morbidity is also vital for patient management. In addition, the finding of this study indicates the necessity of considering schistosomiasis as a cause of the abnormality or as a co-morbidity while interpreting hematological and biochemical profiles in schistosomiasis endemic areas.

## Supporting information

**S1 Questionnaire. Questionnaire for assessing the Hematological and Biochemical changes in *Schistosoma mansoni* infected patients at Haik Primary Hospital, North-East Ethiopia: A comparative cross-sectional study.**
(DOCX)

## Acknowledgments

The authors would like to acknowledge Wollo University for giving support to do this research by proving reagents and equipment used for the laboratory investigation, and Haik Primary Hospital laboratory professional for their help in doing the laboratory tests. We would like also to thank the study participants for their participation in this study.

## Author Contributions

**Conceptualization:** Habtye Bisetegn.

**Data curation:** Habtye Bisetegn, Daniel Getacher Feleke, Habtu Debash, Hussen Ebrahim.

**Formal analysis:** Habtye Bisetegn, Yonas Erkihun.

**Funding acquisition:** Habtye Bisetegn, Daniel Getacher Feleke, Habtu Debash, Hussen Ebrahim.

**Investigation:** Habtye Bisetegn, Daniel Getacher Feleke, Yonas Erkihun.

**Methodology:** Habtye Bisetegn, Hussen Ebrahim.

**Project administration:** Habtye Bisetegn.

**Resources:** Habtye Bisetegn, Habtu Debash, Yonas Erkihun, Hussen Ebrahim.

**Software:** Habtye Bisetegn, Habtu Debash, Yonas Erkihun, Hussen Ebrahim.

**Supervision:** Daniel Getacher Feleke, Habtu Debash, Yonas Erkihun, Hussen Ebrahim.

**Validation:** Habtye Bisetegn.

**Visualization:** Habtye Bisetegn.

**Writing – original draft:** Habtye Bisetegn.

**Writing – review & editing:** Daniel Getacher Feleke, Habtu Debash, Yonas Erkihun, Hussen Ebrahim.

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
