## [Decision Letter · Decision Letter 0]

18 Apr 2022

Dear Mr. Bisetegn,

Thank you very much for submitting your manuscript "Hematological and Biochemical changes in Schistosoma mansoni infected patients at Haik Primary Hospital, North-East Ethiopia: A comparative cross-sectional study" for consideration at PLOS Neglected Tropical Diseases. As with all papers reviewed by the journal, your manuscript was reviewed by members of the editorial board and by several independent reviewers. 

In light of the reviews (below this email and in attachment), we would like to invite the resubmission of a significantly-revised version that takes into account the reviewers' comments. Amongst others, reviewers found that details regarding the sampling procedure and a careful discussion of the limitations and implications of the study are needed.

We cannot make any decision about publication until we have seen the revised manuscript and your response to the reviewers' comments. Your revised manuscript is also likely to be sent to reviewers for further evaluation.

Sincerely,

Alberto Novaes Ramos Jr

Associate Editor

Simone Haeberlein

Deputy Editor

Reviewer's Responses to Questions

**Key Review Criteria Required for Acceptance?**

**Methods**

-Are the objectives of the study clearly articulated with a clear testable hypothesis stated?

-Is the study design appropriate to address the stated objectives?

-Is the population clearly described and appropriate for the hypothesis being tested?

-Is the sample size sufficient to ensure adequate power to address the hypothesis being tested?

-Were correct statistical analysis used to support conclusions?

-Are there concerns about ethical or regulatory requirements being met?

Reviewer #1: Sampling

1. “Systematic random sampling technique was employed to recruit the study participants at Haik Primary Hospital based on the inclusion criteria.” - how was the systematic random sampling conducted? Please elaborate.

2. “According to the rule of thumb recommended by van Voorhis and Morgan, 30 participants per group are required to detect real differences which could lead to about 80% power [21].” - We cannot rely on “rule of thumb”. Please describe appropriate sampling procedure. If random sampling from an adequate sample size was not done, then this should be described in limitations. This would not also be probability sampling

Methods/data collection

1. Both direct fecal smear and KK were used, but only KK was only used in quantification of ova. KK has higher sensitivity than DFS so please explain why KK was not used in determining if a sample is positive.

2. “A blood sample was collected from each study participant and healthy controls following standard operating procedures (SOPs) by trained laboratory technologists” – What are these SOPs? Please describe or at least put a reference or supplementary information.

3. Pictures of the processes are recommended

4. Patient-level (anonymised) were not provided

5. Please include the tools (e.g., questionnaires) used.

Ethical considerations

1. Please provide documentation of ethics clearance - include clearance number and attach clearance

Financial disclosure

1. The study involved fieldwork and specimen processing, so it is strange that the authors declared not receiving funding. Please elaborate how the project expenses (whether in cash or in kind) were funded.

Reviewer #2: The study objectives are well articulated and the study design is appropriate. The study population and the control groups are matching. The sample size is sufficient to draw conclusion and appropriate statistical method is used. Ethical considerations are clearly followed.

Reviewer #3: Yes, the proposed objectives are in accordance with the suggested hypothesis;

Yes, the study design is in line with the objectives;

The population is adequate, however, it lacks a better description of the region of origin of this population in relation to schistosomiasis related to endemicity;

Yes, the sample size is sufficient, however, it needs to better describe the control group;

Yes, the statistical analysis are corrects;

It's okay with ethical requirements.

Reviewer #4: Study Design and Methods:

Study design or procedure: The description of the stages of the study was not shown. For instance, biological samples ( stool and blood) were collected. Did the individuals undergo sample collection at any time during patient assistance at the primary hospital or only at the admission of the study? No details were informed.

Ethical statement was not included in ” Methods”.

“Thus, a total of 180 (90 S. mansoni infected adults and 90 apparently healthy controls) were enrolled in the study”. There was no definition of “apparently healthy controls” described at “Eligibility criteria”. Did any of the controls have a previous history of treated schistosomiasis? If yes, how long ago did they get the medication and proved treated before they were enrolled in the study? 

Since other parasitic infections associated with anaemia may be a confounder, did the author excluded them before enrolling in the study? Is malaria an important issue in the study area? If yes, were both S.mansoni infected and controls investigated for malaria?

**Results**

-Does the analysis presented match the analysis plan?

-Are the results clearly and completely presented?

-Are the figures (Tables, Images) of sufficient quality for clarity?

Reviewer #1: 1. Please describe how the data sets meet the requirements/assumptions for Kruskall-Wallis

2. Please include CIs in all estimates.

Reviewer #2: The presented result matches the analysis plan. and completely presented. comments for consideration are indicated in the last section.

Reviewer #3: Results are well presented, tables and graphs are clear, and statistical tests are well used.

I suggest adding the standard deviation to the graphs.

Reviewer #4: The data were presented mostly in tables that should be reviewed.

Tables and Figures:

No footnotes were added to the respective Tables. Abbreviations were available only at the end of the text. It should be included in the footnotes under each table.

In Table 2 and 3, "normal", "Low" and "High" were not defined ( laboratory reference values should be displayed in the footnote). 

Figure 1 showed a title without legend. Also, there is no identification of the ordinate ( does it represent percentage?). It is advisable to properly review it.

Results: 

In "Intensity of infection, biochemical and hematological values": .....The WBC and RBC count was significantly lower in patients with moderate and heavy intensity of infection compared to patients with light intensity of infection and apparently healthy controls". The results described were not shown.( Table? statistical analysis?)

Discussion:

An extensive literature review could improve the discussion of some issues. Example:

"Erythrocytic sedimentation rate (ESR) was significantly higher in S. mansoni infected patients than

healthy controls. This finding was agreed with reports from Nigeria [23]. The elevation of ESR in

S. mansoni patients might be due to blood loose as a result of bleeding during migration of worms

in the intestine and consumption of blood by the worm". Should ESR be used to diagnosis and monitor inflammatory/fibrotic

response? Was ESR elevated equaly in men and women? May other comorbidities be confounders?

**Conclusions**

-Are the conclusions supported by the data presented?

-Are the limitations of analysis clearly described?

-Do the authors discuss how these data can be helpful to advance our understanding of the topic under study?

-Is public health relevance addressed?

Reviewer #1: 1. The implications of the study are not clear. Please elaborate what will happen if we now know that schistosomiasis may result to biochemical changes

2. Please justify how we the conclusions have global relevance given the limited sample size.

Reviewer #2: Conclusion and recommendation supports the data presented. Authors had discussed the data is helpful and showed public health relevance.

Reviewer #3: The results support the conclusions, however limitations of the study were not described. Although the data are useful for a better understanding of the subject, there was no approach of this importance in relation to public health.

Reviewer #4: The use of hematological and biochemical markers in schistosomiasis management should be done in parallel wirh other tests ( Image). For instance, intensity and extension of fibrotic liver response correlates with progression to chronic and advanced schistosomiasis. Although, authors propose the use of hematological and biochemical markers for screening the individuals infected with Schistosoma, the limitations of the this strategy were not fully discussed. The conclusions are weakly supported by the results. Also, added value of the study and implications of the results in the field were described.

**Editorial and Data Presentation Modifications?**

Reviewer #1: Formatting

1. Include site map

2. Use indentation

3. Please use line numbers

Technical writing

1. Please elaborate what “Institution-based comparative cross-sectional study” means. Not a term I often encounter.

2. “regular chat chewer” - What is chat? Please describe.

3. Please improve English writing and proofread

3.1. “S. mansoni infected adults (≥18 years old) attending Haik Primary Hospital” - people do not attend hospitals

3.2. It’s median, not “media”

4. “Three species of the genus Schistosoma (Schistosoma mansoni, Schistosoma hematobium, and Schistosoma japonicum) are responsible for the majority of Schistosomiasis in the World [2].” – I am not sure if there other genus which causes schistosomiasis?

Reviewer #2: General: all scientific names should be italicized 

Abstract

Conclusion section “Therefore, screening of S. mansoni infected patients for various hematological and biochemical parameters and providing treatment to the underlying abnormalities is very crucial to avoid schistosomiasis associated morbidity and mortality of S. mansoni infected patients.” Should read “Therefore, screening of S. mansoni infected patients for various hematological and biochemical parameters and providing treatment to the underlying abnormalities is very crucial to avoid schistosomiasis associated morbidity and mortality.”

Introduction: Paragraph 3- remove “appendix” 

Method section: since the sociodemographic characters collected are few, delete “Structured” in the questionnaire.

Statistical analysis: “Variables were expressed as median and interquartile range is” repeated 

Result: 

Table 2- show the value for RBC

Under the “Prevalence and severity of anemia among S. mansoni infected patients” section correct “Moreover, 21.4% and 7.14% of females had mild and moderate anemic (Figure 1)” to “…….of females had mild and moderate anemia”

Figure 1 needs revision to show the value for females study subjects and correction of “Non-amenic” to Non-anemic

 Discussion: ESR section instead of “blood loose” it should read “blood loss” 

Confidence interval for intensity of infection should not be repeated in the discussion section.

Reviewer #3: No editorial suggestions

Reviewer #4: The data presentation should be completely reviewed ( Tables and Figure).

**Summary and General Comments**

Reviewer #1: The article touches on a topic seldom studied. However, unless the authors could justify the global implications of the study, the article may be better published in a regional journal.

Reviewer #2: The work is important to show healthcare points to be considered in schistosomiasis patients. The manuscript needs minor language editorial.

Reviewer #3: File in attachment.

Reviewer #4: This is a cross-sectional study that addresses both hematological and biochemical Schistosoma mansoni-induced alterations in infected individuals living in endemic área and attending medical assistance at a primary hospital in Haik town ( North-East Ethiopia). The study population includes adult individuals with schistosomiasis and a group of “apparently” healthy individuals as controls. The findings showed anemia and thrombocytopenia as the major laboratory-based disturbances in addition to abnormal biochemical results in Schistosoma infected individuals. Despite the lack of information about this particular geographical area, the results are not exactly new. Hematological and biochemical profiles of Schistosoma infected individuals were recently described in Amhara region in Ethiopia [Dessie et al, 2020]. Dessie et col. describe the hematological and biochemical profile of Schistosoma mansoni – infected individuals living in a study area with similar altitude, population density and schistosomiasis prevalence as the present study [ Dessie N et al. J Trop Med.2020: 4083252. doi: 10.1155/2020/4083252; Bisetegn H et al. Trop Dis Travel Med Vaccines. 2021 ;7(1):30. doi: 10.1186/s40794-021-00156-0]. The importance of the studies on the subject is undeniable. To stablish the profile of Schistosoma-induced disease by using less expensive laboratory markers may help at least in two ways: to clarify aspects of Schistosoma infection pathological responses and improve disease management. The last one is an important achievement in high-moderate endemic areas in low-income countries. However, the present manuscript brings a mild discussion on the main subject of the study wich should be greatly improved.
---

## [Decision Letter · Decision Letter 1]

2 Jul 2022

Dear Mr. Bisetegn,

Thank you very much for submitting a revised version of your manuscript "Hematological and Biochemical changes in Schistosoma mansoni infected patients at Haik Primary Hospital, North-East Ethiopia: A comparative cross-sectional study" for consideration at PLOS Neglected Tropical Diseases. Based on the new reviews, we are likely to accept this manuscript for publication, providing that you modify the manuscript according to the review recommendations. Please pay attention to the reviewer comments below.

Sincerely,

Alberto Novaes Ramos Jr

Associate Editor

Simone Haeberlein

Deputy Editor

Reviewer's Responses to Questions

**Key Review Criteria Required for Acceptance?**

**Methods**

-Are the objectives of the study clearly articulated with a clear testable hypothesis stated?

-Is the study design appropriate to address the stated objectives?

-Is the population clearly described and appropriate for the hypothesis being tested?

-Is the sample size sufficient to ensure adequate power to address the hypothesis being tested?

-Were correct statistical analysis used to support conclusions?

-Are there concerns about ethical or regulatory requirements being met?

Reviewer #1: Can we properly calculate the sample size based on your research question and study design? I am not sure the article in Tutorials in Quantitative Methods for Psychology is an authoritative reference for sample size estimation.

"The quality of the blood sample was maintained by collecting and processing it according to the229

standard operating procedures (SOPs) - the authors said that they had already provided a reference, but there is still none.

Reviewer #2: The objectives of the study are clearly stated and the study design is appropriate to address it. the sample size is appropriate according to WHO standard. The statistical analysis is is appropriate to address the objectives.

Reviewer #3: (No Response)

**Results**

-Does the analysis presented match the analysis plan?

-Are the results clearly and completely presented?

-Are the figures (Tables, Images) of sufficient quality for clarity?

Reviewer #1: (No Response)

Reviewer #2: The results are clear and completely presented with tables.

Reviewer #3: (No Response)

**Conclusions**

-Are the conclusions supported by the data presented?

-Are the limitations of analysis clearly described?

-Do the authors discuss how these data can be helpful to advance our understanding of the topic under study?

-Is public health relevance addressed?

Reviewer #1: (No Response)

Reviewer #2: the conclusion stems from the study findings . There is no limitation of analysis described.. The study finding is helpful to advance the understanding of the topic under study and has public health importance.

Reviewer #3: (No Response)

**Editorial and Data Presentation Modifications?**

Reviewer #1: (No Response)

Reviewer #2: Accept it.

Reviewer #3: (No Response)

**Summary and General Comments**

Reviewer #1: Great work on the revisions. I just have minior comments. To ensure transparency, please provide the anonymised/coded data you used in the analysis. Note that policy of PLOS - "The PLOS Data policy requires authors to make all data underlying the findings described in their manuscript fully available without restriction, except in cases where the data are legally or ethically restricted (for example, participant privacy is an appropriate restriction)" The ethical restriction should not be an issue if the each participant is coded.

Reviewer #2: None

Reviewer #3: 

My main suggestions and recommendations I sent in an attached file. From what I observed in the answers of the authors, they did not access this file. As for the other suggestions made in the general questionnaire, I fully agree.

I am attaching here again the PDF file mentioned above.

PLOS NTD--CONSIDERATIONS-2022.pdfHematological and Biochemical changes in Schistosoma mansoni infected patients at

Haik Primary Hospital, North-East Ethiopia: A comparative cross-sectional study

GENERAL CONSIDERATIONS AND SUGGESTIONS

Methodology

It is necessary to clarify the origin of the control group because it is a little strange to have a control group chosen within a hospital. The criterion used to determine the participants in this group has been based only on clinical information as mentioned in the item Socio-demographic characteristics and clinical data? Stool tests has been done? If yes, which method has been used?

The authors only mention (in the introduction) that schistosomiasis mansoni is endemic in Haik town, not revealing this endemicity index, an important factor in the analysis and determination of the groups.

A more detailed description of the area from which these patients originated was lacking. It is important to know if the region has S. haematobium, for example, and if this was also used as an exclusion criterion.

In the Item “Stool sample collection and processing”, the last period about blood collection must be removed and added in the following item – “Blood sample collection processing”

Patients with splenomegaly were excluded and those with hepatomegaly, too? If these conditions are present in patients with severe forms of schistosomiasis, wouldn't it be interesting to keep them in the study group? An association could be made between the presence of these clinical forms and the hematological and biochemical indices analyzed.

Results.

The results are clear and well-presented between tables and graphs, I suggest, however, that you add the standard deviation in the graphs.

Discussion

The discussion in relation to hematological and biochemical indices is well founded, however I missed the approach on clinical and socio-demographic data.

There was a lack of an approach to public health using indicators from the region and also, the limitations found.

PLOS authors have the option to publish the peer review history of their article (what does this mean?). If published, this will include your full peer review and any attached files.

Reviewer #1: No

Reviewer #2: No

Reviewer #3: No

Figure Files:

Data Requirements:

Reproducibility:

References

---

## [Decision Letter · Decision Letter 2]

10 Aug 2022

Dear Mr. Bisetegn,

We are pleased to inform you that your manuscript 'Hematological and Biochemical changes in Schistosoma mansoni infected patients at Haik Primary Hospital, North-East Ethiopia: A comparative cross-sectional study' has been provisionally accepted for publication in PLOS Neglected Tropical Diseases.

Before your manuscript can be formally accepted you will need to complete some formatting changes, which you will receive in a follow up email. A member of our team will be in touch with a set of requests. Please also pay attention to the requested Editorial and Data Presentation Modifications raised by reviewer 2, shown below this email.

Sincerely,

Alberto Novaes Ramos Jr

Academic Editor

Simone Haeberlein

Section Editor

Reviewer's Responses to Questions

**Key Review Criteria Required for Acceptance?**

**Methods**

-Are the objectives of the study clearly articulated with a clear testable hypothesis stated?

-Is the study design appropriate to address the stated objectives?

-Is the population clearly described and appropriate for the hypothesis being tested?

-Is the sample size sufficient to ensure adequate power to address the hypothesis being tested?

-Were correct statistical analysis used to support conclusions?

-Are there concerns about ethical or regulatory requirements being met?

Reviewer #1: (No Response)

Reviewer #2: The objectives of the study are clearly presented. The study design and the sample size is appropriate to evaluate the hypothesis. Correct statistical methods are used to analyze the data to generate conclusion. The work is done following ethical considerations.

Reviewer #3: The objective, study design, sample size and statistical analysis with corrections to the suggestions made are in accordance with the purpose of the manuscript.

**Results**

-Does the analysis presented match the analysis plan?

-Are the results clearly and completely presented?

-Are the figures (Tables, Images) of sufficient quality for clarity?

Reviewer #1: (No Response)

Reviewer #2: The analysis presented match with the analysis plan. All results are presented according to the objectives of the study. Figures and Tables are of good quality.

Reviewer #3: The results are well presented and better visualized with the present figures. The tables are well made.

**Conclusions**

-Are the conclusions supported by the data presented?

-Are the limitations of analysis clearly described?

-Do the authors discuss how these data can be helpful to advance our understanding of the topic under study?

-Is public health relevance addressed?

Reviewer #1: (No Response)

Reviewer #2: The conclusion is supported by the data obtained from the study. Limitations are also addressed. Moreover, it had shown the public health importance.

Reviewer #3: The conclusions support the data presented. The authors report the limitations of the work, however I still think there is still a need for a better discussion of the impact of these results in the region using the local public health indicators as parameters.

**Editorial and Data Presentation Modifications?**

Reviewer #1: (No Response)

Reviewer #2: The MS PDF is presented in double copies.

Author summary first sentence needs rewriting. Introduction line 80 83 should be rewritten clearly through avoiding long sentence. In the method section the geographical location should be put in correct description. Line 136 needs rewriting. Line number 290 add 'for' after values. Line334 "health" should read 'healthy". The description of Line 340-347 is not supported by the current study outcome. Hence, needs revision.

Reviewer #3: No suggestion.

**Summary and General Comments**

Reviewer #1: Authors should submit a tracked changes version of manuscript or a version where only the changes are highlighted

Patient-level data (anonymised) should be provided. I think it is a requirement that data used in coming up with the results should be available

Responses to the comments should have been incorporated in the text. For instance, the following response should be incorporated in the manuscript.

Response: Institution-based mean the study is conducted in health facilities/institution/ and it is

not community based/survey. While comparative crossectional study is a a form of cross-sectional

study where data on the two groups (in our case S. mansoni infected patients and healthy control)

are collected at a point on time

Response: The variables are continuous variables. The distributions are not normally distributed

as checked by Kolmogorov-Smirnov and Shapiro Wilk normality tests and the test should be non-

parametric. It is four group (patient with light intensity, moderate intensity, heavy intensity of

infection and healthy controls). If the number of groups to be compared are three and more, the

Kruskal-Wallis H test is used to compare the variable.

Reviewer #2: The work has significant value as a public health concern. It needs minor grammatical revision and italicizing scientific names.

Reviewer #3: I understand that with the modifications made by the reviewers, the article is more robust and that the editor's discretion can follow for publication.

PLOS authors have the option to publish the peer review history of their article (what does this mean?). If published, this will include your full peer review and any attached files.

Reviewer #1: No

Reviewer #2: No

Reviewer #3: No

---

## [Editor Report · Acceptance letter]

25 Aug 2022

Dear Mr. Bisetegn,

We are delighted to inform you that your manuscript, "Hematological and Biochemical changes in Schistosoma mansoni infected patients at Haik Primary Hospital, North-East Ethiopia: A comparative cross-sectional study," has been formally accepted for publication in PLOS Neglected Tropical Diseases.

Best regards,

Shaden Kamhawi

co-Editor-in-Chief

Paul Brindley

co-Editor-in-Chief
